



# AMPERE Polar Cap Boundaries

Angeline G. Burrell[1], Gareth Chisham[2], Stephen E. Milan[3], Liam Kilcommons[4], Yun-Ju Chen[5], Evan
G. Thomas[6], and Brian Anderson[7]

[1]U.S. Naval Research Laboratory, 4555 Overlook Ave SW, Washington, DC, USA
[2]British Antarctic Survey, Cambridge, UK
[3]University of Leicester, University Road, Leicester, UK
[4]University of Colorado, Boulder, 2055 Regent Drive, Boulder, CO, USA
[5]University of Texas at Dallas, 800 West Campbell Road, Richardson, TX, USA
[6]Dartmouth College, 14 Engineering Drive, Hanover, NH, USA
[7]Johns Hopkins University Applied Physics Laboratory, 11100 Johns Hopkins Road, Laurel, MD, USA

**Correspondence:** A.G. Burrell (angeline.burrell@nrl.navy.mil)

**Abstract.** The high latitude atmosphere is a dynamic region with processes that respond to forcing from the Sun, magnetosphere, neutral atmosphere, and ionosphere. Historically, the dominance of magnetosphere-ionosphere interactions has motivated upper atmospheric studies to use magnetic coordinates when examining magnetosphere-ionosphere-thermosphere coupling processes. However, there are significant differences between the dominant interactions within the polar cap, auroral oval,

and equatorward of the auroral oval. Organising data relative to these boundaries has been shown to improve climatological and statistical studies, but the process of doing so is complicated by the shifting nature of the auroral oval and the difficulty in measuring its poleward and equatorward boundaries.

This study presents a new set of open-closed magnetic field line boundaries (OCBs) obtained from Active Magnetosphere and Planetary Electrodynamics Response Experiment (AMPERE) magnetic perturbation data. AMPERE observations of field

aligned currents (FACs) are used to determine the location of the boundary between the Region 1 (R1) and Region 2 (R2) FAC systems. This current boundary is thought to typically lie a few degrees equatorward of the OCB, making it a good candidate for obtaining OCB locations. The AMPERE R1/R2 boundaries are compared to the Defense Meteorological Satellites Program Special Sensor J (DMSP SSJ) electron energy flux boundaries to test this hypothesis and determine the best estimate of the systematic offset between the R1/R2 boundary and the OCB as a function of magnetic local time. These calibrated boundaries,

as well as OCBs obtained from Magnetopause-to-Aurora Global Exploration (IMAGE) observations, are validated using simultaneous observations of the convection reversal boundary measured by DMSP. The validation shows that the OCBs from IMAGE and AMPERE may be used together in statistical studies, providing the basis of a long-term data set that can be used to separate observations originating inside and outside of the polar cap.



# 1 Introduction

The high latitude atmosphere is a dynamic region driven by solar and magnetospheric forcing. The dominant coupling occurs between the ionosphere and magnetosphere, which drives plasma motions in the auroral oval and polar cap through the Dungey cycle (Dungey, 1961). These motions differ based on whether the ionospheric plasma lies on open or closed geomagnetic

field lines, where open field lines are those that reach out from the Earth to connect with the Interplanetary Magnetic Field (IMF) and closed field lines connect back to the Earth in the opposite hemisphere. In the simplest case, convective drifts within the polar cap ionosphere travel along approximately straight, antisunward paths (from magnetic local noon to midnight) and convective drifts in the auroral oval travel in curved, sunward paths. The auroral and polar cap regions also experience different types of magnetosphere-ionosphere-thermosphere (MIT) coupling. For example, field-aligned currents (FACs) flow

between the ionosphere and the magnetosphere at auroral latitudes (Coxon et al., 2018, and references therein). In the polar cap, the antisunward ionospheric convection flow driven by magnetic reconnection on the Earth's dayside magnetopause causes a highly structured polar ionosphere as the dense dayside ionospheric plasma is transported to the nightside where recombination processes destroy plasma that is not returned to sunlit regions quickly enough (e.g., Spiro et al., 1978). Focusing on the auroral oval, the high rate of particle precipitation in this region leads to additional Joule heating in the thermosphere (e.g., Vasyliūnas

and Song, 2005).

Due to the differences in ionospheric and thermospheric behavior in the auroral oval and the polar cap, it is desirable to have a coordinate system that indicates in which region measurements were taken. This type of adaptive, high-latitude gridding has been performed with various data sets (Redmon et al., 2010; Chisham, 2017b; Kilcommons et al., 2017). These studies have demonstrated improved statistical and climatological results when using adaptive, high-latitude coordinates. Unfortunately,

observations of the open-closed magnetic field line boundary (OCB) are sparse. Long-term and large-scale studies would benefit from specifications of the OCB in both hemispheres and all magnetic local times (MLTs) every 15 min or less (Cowley and Lockwood, 1992). Models that have the ability to distinguish between regions with open and closed field lines would also benefit from adaptive, high-latitude coordinates (Zhu et al., 2019).

This study presents a new set of OCBs obtained from the Active Magnetosphere and Planetary Electrodynamics Response

Experiment (AMPERE) magnetic perturbation observations. AMPERE measurements of FACs make it possible to estimate the location where Region 1 (R1) and Region 2 (R2) FAC systems meet (the R1/R2 boundary). Because the location of the Birkeland current system is tied to the OCB, it seems logical to hypothesize that a dependable relationship between the R1/R2 boundary and the OCB exists. This study investigates the relationship between the AMPERE R1/R2 boundary and the OCB measured by the Defense Meteorological Satellites Program Special Sensor J (DMSP SSJ) electron energy flux boundaries.

Section 2 presents the details of both data sets. Section 3 explores the relationship between the different boundaries and presents the calibration process that allows the AMPERE R1/R2 boundary to be used as a proxy for the OCB. This calibration, as well as the previous Magnetopause-to-Aurora Global Exploration (IMAGE) calibration performed by Chisham (2017b), is validated in section 4 by comparing calibrated OCBs with the convection reversal boundaries (CRBs) from DMSP plasma drift measurements and summarized in section 5.





## 2   Instrumentation

The data sets used in this study have a long and ongoing history of observations. The primary data set, AMPERE, is described in section 2.1. Two instruments from DMSP are used, one for calibration of the boundaries and another for validation. Both DMSP data sets are described in section 2.2. The IMAGE far ultraviolet (FUV) data set used in the validation is described in
section 2.3.

### 2.1   AMPERE

AMPERE assimilates measurements from the approximately 70 polar-orbiting spacecraft of the Iridium telecommunications constellation to deduce the high-latitude distribution of horizontal magnetic field perturbations produced by the FACs responsible for magnetosphere-ionosphere coupling (Anderson et al., 2000, 2002; Waters et al., 2001; Coxon et al., 2018). The FAC
pattern in both hemispheres is calculated from 10-minute averages at a 2 min cadence on a magnetic latitude and MLT grid ($1°$ $\times 1$ h resolution); this study employs observations from 2010–2012.

### 2.2   DMSP

The DMSP OCB locations are obtained from energetic electron fluxes measured by three DMSP spacecraft (F16-F18) that were operational and have updated ephemera (Redmon et al., 2017) during the period of time when AMPERE R1/R2 boundaries
were available. The DMSP satellites were located in sun-synchronous polar orbits at an altitude of about 830 km, with an orbital period of approximately 101 min. The geographic locations of the DMSP SSJ/5 equatorward and poleward boundaries were determined using ssj_auroral_boundary (Kilcommons and Burrell, 2019), which implements the technique described in Kilcommons et al. (2017). A clean set of OCBs were obtained by selecting the poleward boundaries with figures of merit greater than 3.0 and calculating the AACGM-v2 coordinates at each location (Shepherd, 2014; Burrell et al., 2018b).
The same DMSP spacecraft also carry an Ion Velocity Meter (IVM) that measures the three dimensional ion velocity (Heelis and Hanson, 1998). Because the convective plasma drifts are strongly tied to the motion and state of the magnetic field lines, the CRB is typically located at or just equatorward of the OCB (Newell et al., 2004; Drake et al., 2009) except for regions of the dayside and nightside ionosphere that map to regions of ongoing magnetic reconnection. The CRB is the location where plasma drifts change from moving sunward to antisunward, or vice versa.
In this paper, CRBs obtained by Chen et al. (2015) are used to validate the AMPERE OCB locations within an hour of dawn (06:00 MLT) and dusk (18:00 MLT). Other MLTs were not considered for several reasons. Most importantly:

1. Near magnetic noon and midnight the flows tend to be mostly sunward or antisunward, meaning there is no clear reversal in the convection as a function of magnetic latitude.

2. The IMF orientation will shift the MLT location of these sunward or antisunward flows, meaning more local times than
just noon and midnight are affected.

3. Near midnight, the Harang reversal can give the appearance of multiple convection reversals at different latitudes.





The Chen et al. (2015) algorithm is optimized to identify the CRB in a two-cell convection pattern. If the plasma convection has a complex pattern with more than four reversals, or the plasma flows are weak and noisy, the program will not identify any CRB location. For symmetric, multi-cell patterns (such as those observed when the IMF is dominated by a positive $B_Z$ component), the program will identify the most equatorward reversal boundary. Otherwise, the most poleward reversal boundary

will be selected as the CRB location. The algorithm typically performs better in the summer, since the DMSP IVM performs better when the plasma density is higher (Chen et al., 2015; Chen and Heelis, 2018).

## 2.3 IMAGE FUV

Chisham (2017b) obtained estimates of the OCB from auroral images measured by the FUV imagers onboard the IMAGE spacecraft. Images of the northern hemisphere auroral region were available for the epoch spanning May 2000 to August 2002.

During this time, the spacecraft was located in an elliptical orbit with a 90° inclination, an apogee of 7 $R_E$, a perigee of 1000 km, and an orbital period of ∼13.5 h.

This study uses data from the two FUV spectographic imagers, SI12 and SI13 (Mende et al., 2000). The SI13 imager measured oxygen emissions at 135.6 nm, resulting from energetic electron precipitation. The SI12 imager measured Doppler-shifted Lyman-$\alpha$ emissions at 121.8 nm, resulting from proton precipitation. Both imagers provided data at a 2 min resolution,

when the northern hemisphere is visible. The OCB was identified in the individual FUV images and fit across all magnetic local times using the techniques described by Longden et al. (2010) and Chisham (2017b).

## 3 Relationship between the R1/R2 boundary and OCB

This study follows the process outlined in Boakes et al. (2008), which determined the offset between the IMAGE FUV poleward auroral boundaries and DMSP OCBs, to obtain a correction between the AMPERE R1/R2 boundary and the DMSP SSJ OCBs.

The five steps of this process are enumerated below.

1. Identify the AMPERE R1/R2 boundaries.

2. Pair AMPERE R1/R2 boundaries with DMSP SSJ OCBs.

3. Determine the typical offset at different MLTs.

4. Find a functional fit that describes the offset between the DMSP SSJ OCBs and the AMPERE R1/R2 boundaries.

5. Use the functional fit to correct the AMPERE R1/R2 boundary locations, creating an AMPERE OCB proxy.

The basis of the R1/R2 boundary identification is a fitting technique described by Milan et al. (2015). This technique aims to determine the centre and radius of the circle that best describes the boundary between the R1 and R2 FACs that were first identified by Iijima and Potemra (1976) without fitting to individual MLT bins. By avoiding this common method of defining a high-latitude boundary, this R1/R2 boundary identification is more robust in the event of sparse or weak currents and less

influenced by the poorly defined current structures near local magnetic noon and midnight.





The following procedure is applied to each AMPERE FAC grid. In this description, positive and negative values represent upward and downward currents, respectively. The R1 currents flow upwards at dusk and downwards at dawn, while the R2 currents have the opposite polarity and lie equatorward of the R1 current system. To distinguish between these two FAC systems, the first step is to multiply all FAC magnitudes on the dawn side ($00:00 \leq$ MLT $< 12:00$) by -1. This redefines the current signs such that R1 FACs are positive and R2 FACs are negative at all MLTs. Then a center point ($x_0$, $y_0$) is assumed, where $x_0$ is the dawnward distance from the noon-midnight meridian and $y_0$ is the sunward distance from the dawn-dusk meridian. A range of centres are tested, with $x_0$ varying between $\pm 4°$ and $y_0$ varying between -6° and 0° latitude. Additionally, a range of radii are tested at each centre point; varying the radius by 1° latitude (111 km) from 8° to 35°. At each radius and centre point the sum of the FACs at 200 equally-spaced points in a ring centred at ($x_0$, $y_0$) is found. This produces a profile of integrated FAC magnitude with radius, in which a negative-positive bipolar signature is sought. The zero-crossing of the bipolar signature is taken to be the R1/R2 boundary and the peak-to-peak magnitude provides a figure of merit (FOM) for the boundary fit. For each AMPERE FAC grid, the circle with the best FOM is chosen and grids with low FOMs are discarded as being unreliable.

This study uses AMPERE R1/R2 boundaries made from January 2010 through December 2012. Using only R1/R2 boundaries with FOMs greater than 0.15 mA provides 636,250 northern and 531,666 southern hemisphere boundary locations. Pairing these boundaries to good DMSP SSJ OCB detections by requiring each observation be taken within 10 min of each other leaves 29,683 northern and 29,135 southern hemisphere boundaries. Good DMSP SSJ OCB detections are defined as having a FOM of 3.0 or greater. This is consistent with the work presented by Kilcommons et al. (2017) and reduces the number of passes with dayside precipitation associated with the cusp, mantle, and other sources whose origin (inside or outside the polar cap) is still debatable. The DMSP SSJ paired OCBs for each hemisphere and satellite are shown in Figure 1 as a scatter plot, with the median location of the AMPERE R1/R2 boundaries plotted on top. Note that the R1/R2 boundaries lie near the equatorward edge of the DMSP SSJ OCBs. Because of the DMSP satellite orbits, MLTs near noon are only covered in the northern hemisphere and those near midnight are covered only in the southern hemisphere.

Ideally, observations from both hemispheres can be combined to provide complete MLT coverage of the differences between the AMPERE R1/R2 boundaries and DMSP SSJ OCBs. To test the assumption that the northern and southern boundaries have the same local time dependence, the MLT bins with observations in both hemispheres (05:00-08:00 and 15:00-20:00 MLT) were compared. The hourly boundary offsets in each hemisphere and both hemispheres combined, all calculated using the magnetic co-latitude, are presented in Table 1.

The boundary offsets in Table 1 were calculated by finding the typical difference between the DMSP SSJ OCB and the AMPERE R1/R2 boundary location in AACGM-v2 magnetic latitude in one hour MLT bins. The typical boundary latitude difference ($\Delta\phi$, which equals the DMPS SSJ OCB co-latitude minus the AMPERE R1/R2 boundary co-latitude) is represented by two values, the median of the boundary latitude differences and the peak of a Gaussian distribution (S.G. peak), fitted to a smoothed histogram (as in Boakes et al., 2008). The histograms have 1° bins, and were smoothed using a 4° running average. The smoothed histogram was then fit with a Gaussian function, allowing the S.G. peak and standard deviation to be calculated.





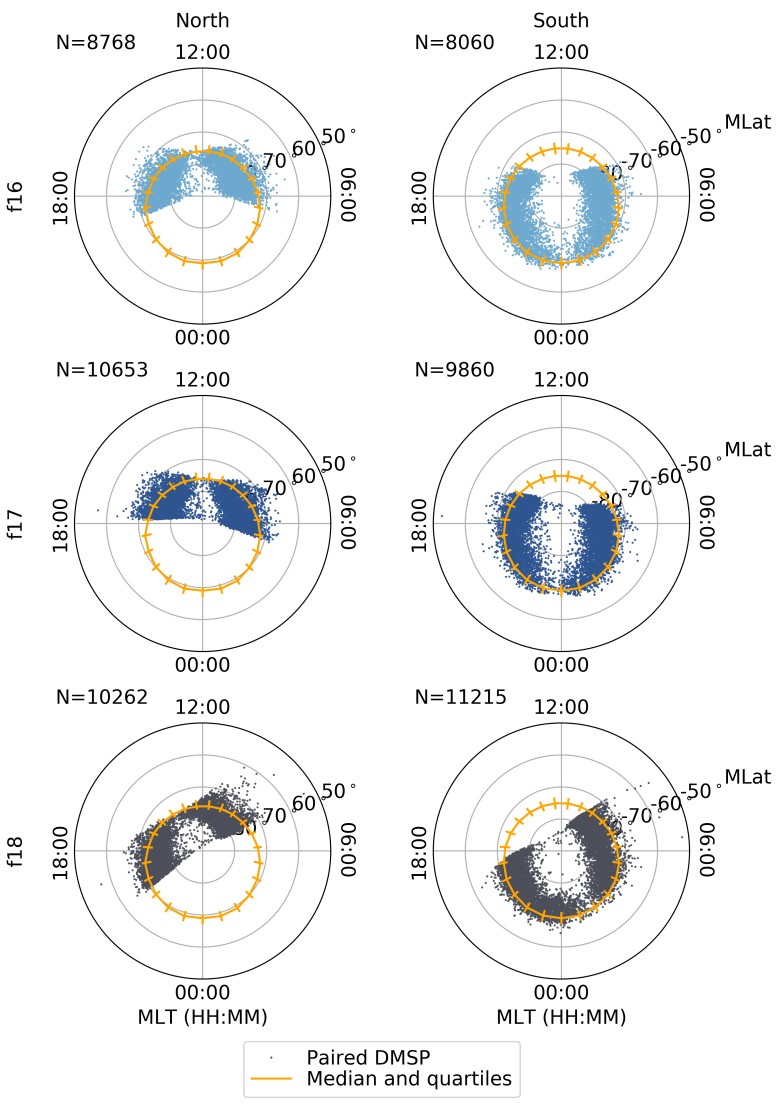

**Figure 1.** Paired AMPERE R1/R2 boundaries and DMSP SSJ OCBs for both hemispheres (northern in the left column and southern in the right column) and each satellite. The scattered points show the DMSP SSJ OCBs, while the gold circle shows the median location of the AMPERE R1/R2 boundaries. The scatter bars denote the quartiles of the paired AMPERE R1/R2 boundaries.





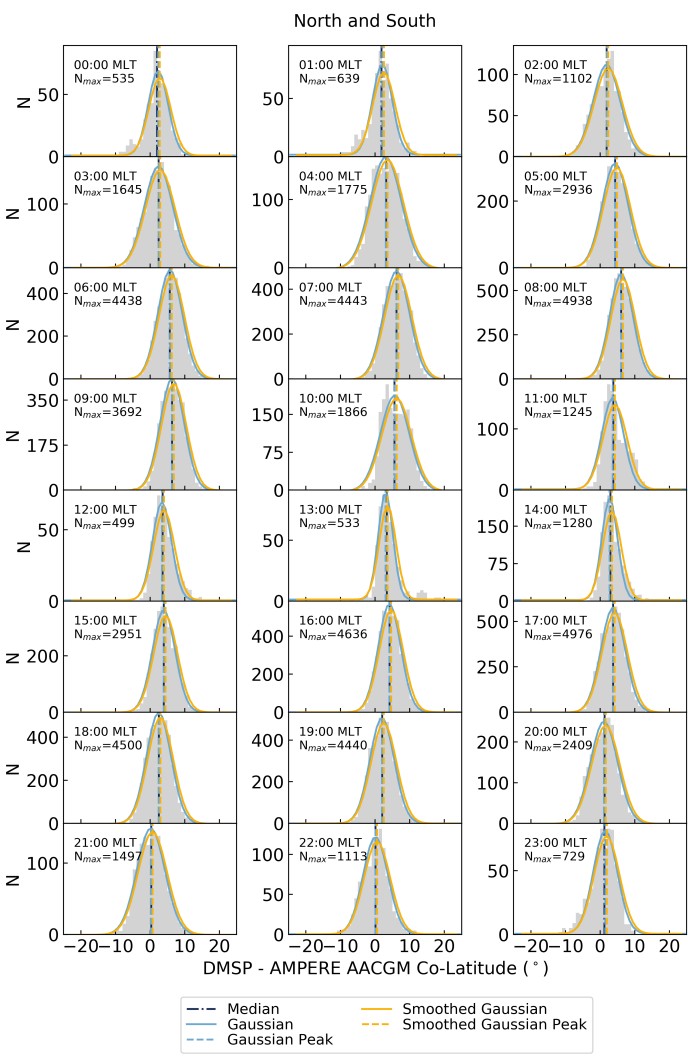

**Figure 2.** Hourly distributions of paired AMPERE R1/R2 boundary and DMSP SSJ OCB latitude differences, with boundary differences from both hemispheres and all satellites. The black dashed line shows the median of the distribution, the blue line shows a Gaussian fit to the distribution, and the gold line shows the Gaussian fit to the smoothed histogram. The vertical blue and gold lines show the peaks of each Gaussian fit.





**Table 1.** Hourly boundary offset for hours with over 100 boundary pairs and successfully fit Gaussians

| MLT | North | | South | | Both | |
|---|---|---|---|---|---|---|
| | Median (°) | S.G. Peak (°) | Median (°) | S.G. Peak (°) | Median (°) | S.G. Peak (°) |
| 00:00 | - | - | 2.04 | 2.83 | 2.04 | 2.83 |
| 01:00 | - | - | 1.88 | 2.56 | 1.88 | 2.56 |
| 02:00 | - | - | 1.93 | 2.36 | 1.93 | 2.36 |
| 03:00 | - | - | 2.46 | 2.94 | 2.46 | 2.94 |
| 04:00 | - | - | 3.20 | 3.60 | 3.20 | 3.60 |
| 05:00 | 3.96 | 4.45 | 4.80 | 5.29 | 4.33 | 4.86 |
| 06:00 | 5.16 | 5.69 | 6.34 | - | 5.73 | 6.26 |
| 07:00 | 5.29 | 5.88 | 6.98 | - | 6.21 | 6.71 |
| 08:00 | 5.69 | 6.19 | 7.10 | - | 6.08 | 6.64 |
| 09:00 | 5.38 | 5.99 | - | - | 6.35 | 6.88 |
| 10:00 | 4.64 | 5.29 | - | - | 5.64 | 6.23 |
| 11:00 | 3.78 | 4.27 | - | - | 3.82 | 4.32 |
| 12:00 | 3.57 | 3.99 | - | - | 3.66 | 4.04 |
| 13:00 | 3.30 | 3.61 | - | - | 3.40 | 3.62 |
| 14:00 | 2.95 | 3.36 | - | - | 3.02 | 3.43 |
| 15:00 | 3.49 | 3.97 | 5.21 | 5.76 | 3.97 | 4.50 |
| 16:00 | 4.20 | 4.68 | 4.19 | 4.66 | 4.19 | 4.67 |
| 17:00 | 4.00 | 4.47 | 3.32 | 3.74 | 3.77 | 4.22 |
| 18:00 | 2.82 | 3.30 | 2.27 | 2.77 | 2.54 | 3.01 |
| 19:00 | 2.67 | 3.12 | 1.52 | 1.95 | 2.07 | 2.51 |
| 20:00 | 2.42 | 3.13 | 0.96 | 1.35 | 1.29 | 1.63 |
| 21:00 | - | - | 0.33 | 0.73 | 0.33 | 0.73 |
| 22:00 | - | - | 0.14 | 0.60 | 0.14 | 0.60 |
| 23:00 | - | - | 1.24 | 1.94 | 1.24 | 1.94 |

Comparing the median and S.G. peak of the $\Delta\phi$ for the MLT bins with observations in both hemispheres shows a mean hemispheric difference of -0.30° and 0.23° for the median and S.G. peaks, respectively. This difference is small enough to justify combining the northern and southern hemispheric $\Delta\phi$, since it is much smaller than the mean standard deviation of the MLT distributions ($\bar{\sigma} = 2.66°$ for the overlapping MLT bins). The results for the combined hemispheres are presented in the rightmost columns of Table 1 and in Figure 2. There is about a 0.49° difference between the median and S.G. peak values. This difference is very small compared to the width of the $\Delta\phi$ distributions, and provides a measure of uncertainty for the resulting boundary correction.



**Table 2.** Boundary fit constants for DMSP-AMPERE boundary offset

| Constant | Median | S.G. Peak |
|----------|--------|-----------|
| $a$ | $4.01°$ | $4.41°$ |
| $e$ | -0.55 | -0.51 |
| $\tau$ | -0.92 | -0.95 |

Unfortunately, the differences between the boundary fitting methodology used by Chisham (2017b) and Milan et al. (2015) mean that it is not reasonable to use a harmonic function to describe the offset between the DMSP SSJ OCBs and the AMPERE R1/R2 boundaries, as done in prior auroral boundary fitting studies (R.H. Holzworth, 1975; Carbary et al., 2003; Boakes et al., 2008). Because the R1/R2 boundary fitting method used by Milan et al. (2015) does not fit a series of MLT bins, the boundary

correction cannot be applied prior to circle fitting and will determine the final shape of the OCB proxy. Thus, this study uses a generalised ellipse (equation 1) rather than a harmonic function to avoid overfitting the MLT dependence of the offset between the DMSP SSJ OCBs and the AMPERE R1/R2 boundaries.

$$K(\lambda) = \frac{a\left(1 - e^2\right)}{1 + e\cos\left(\lambda - \tau\right)} \tag{1}$$

In equation 1, $\lambda$ is the MLT in radians, $a$ is the semi-major axis in degrees, $e$ is the eccentricity (a unitless quantity), and $\tau$

is the angular offset of the ellipse's centre in radians. These four constants allow the ellipse to adjust its centre and axes. They are fit using the Python SciPy least squares fitting routine, *leastsq* (Jones et al., 2001–), which wraps the MINPACK *lmdif* and *lmder* algorithms (More et al., 1984). The least squares fitting routine minimises the difference between $K$ and $\Delta\phi$, weighted by the inverse of the error, $\epsilon$. The error is defined as shown in equation 2, where $N_{MLT}$ is the number of $\Delta\phi$ observations in each MLT bin, $N_{max}$ is the maximum $N_{MLT}$, and $\sigma$ is either the interquartile range or the standard deviation depending on

whether the median or S.G. peak was used as the central value. The results of this fitting procedure are shown in Figure 3 and Table 2.

$$\epsilon = \sqrt{\left(\frac{N_{MLT}}{N_{max}}\right)^2 + \sigma^2} \tag{2}$$

As shown in Figure 3, the AMPERE R1/R2 boundary lies about $2°$ equatorward of the OCB at magnetic midnight, about $4°$ equatorward of the OCB at magnetic noon, and further out at dawn and dusk. The elliptical fit follows the central values

very closely between 00:00 and 10:00 MLT, and smooths through the maxima and minima at 12:00, 16:00, and 22:00 MLT. Even where the differences are greatest, though, the elliptical fit does not differ from the central value by more than $\frac{\epsilon}{2}$. This behaviour is consistent whether the median or S.G. peak is used in the fitting process. Indeed, the semi-major axis differs by less than the typical difference between the median and S.G. peak values and the eccentricity and angular offset are even more similar.



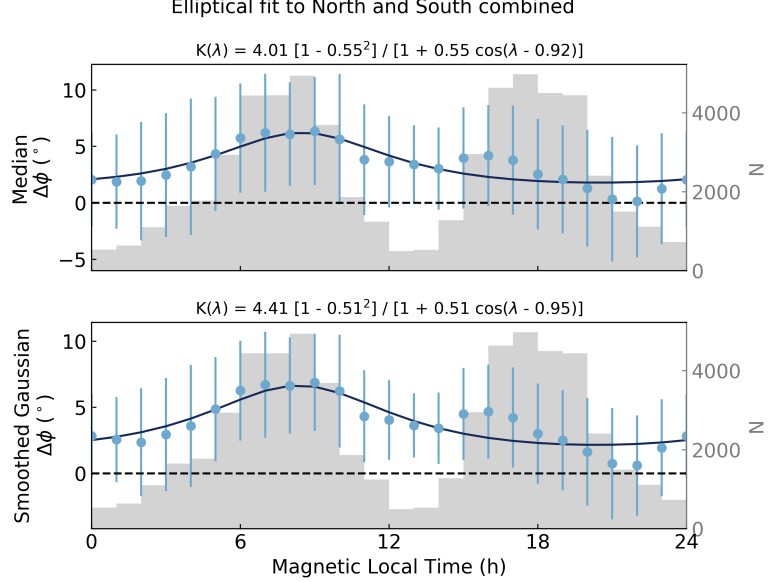

**Figure 3.** Elliptical boundary correction (black line) fit to the median (top) and S.G. peak (bottom) $\Delta\phi$ for both hemispheres. The blue dots and scatter bars show the central value and $\epsilon$ in each MLT bin. The grey histogram shows $N_{MLT}$, and scales to the y-axis on the right.

The consistency of the elliptical fit for both central values, as well as its success at capturing the major features of $\Delta\phi$ given the functional constraints, make it a good candidate for correcting the R1/R2 boundary to provide an OCB estimate. The Gaussian nature of the hourly bins (shown in Figure 2) suggests that differences between the R1/R2 boundary and DMSP SSJ OCB are randomly distributed, confirming the conclusion that it is appropriate to use $K$ to correct the R1/R2 boundary to obtain an AMPERE OCB estimate.

## 4 Validation

The appropriateness of using $K$ to transform the AMPERE R1/R2 boundary into an AMPERE OCB is tested by comparing the AMPERE OCBs to the DMSP CRBs within an hour of dawn and dusk. To ensure that the performance of the AMPERE OCBs are on par with previous OCB calculations, this validation is also performed for the IMAGE OCBs. Unfortunately, it is impossible to directly compare the AMPERE and IMAGE OCBs because there is no temporal overlap between the two data sets. This validation effort paired OCBs with DMSP CRBs that were identified within 10 min of one another. The location of the DMSP CRB relative to the OCB was then determined. In this adaptive coordinate system, the OCB is set at a co-latitude of 74° (a latitude chosen to represent the OCB in adaptive, high-latitude coordinates based on the typical size of the polar cap). CRBs that occur poleward or equatorward of the OCB will have co-latitudes greater than or less than 74°, respectively. This adaptive gridding was performed using the Python package, ocbpy (Burrell and Chisham, 2018; Burrell et al., 2018a).



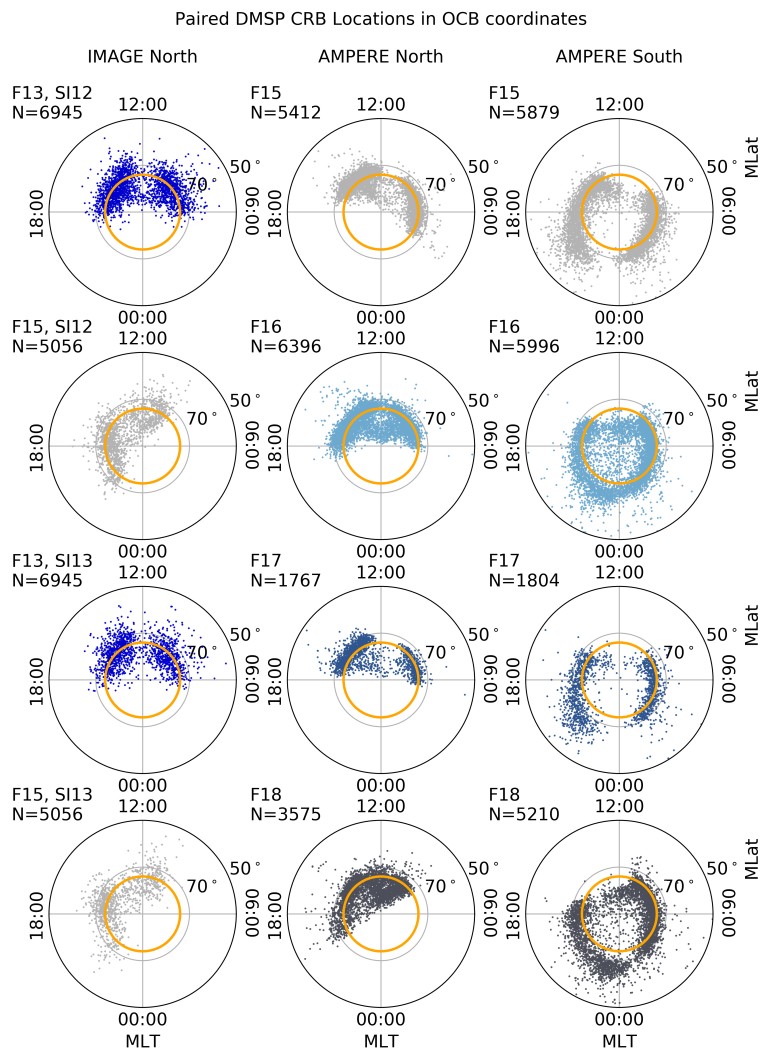

**Figure 4.** Paired IMAGE and AMPERE OCBs with DMSP CRBs for the available hemispheres and each satellite. The IMAGE data shows the SI-12 and SI-13 observations for the northern hemisphere (left column), while the median elliptical correction was applied to obtain the AMPERE OCBs shown in the middle and right columns (which show the northern and southern hemispheres, respectively). The scattered points show the DMSP IVM CRBs, while the gold circle shows the IMAGE or AMPERE OCB. To simplify the comparison, the DMSP IVM CRB locations are plotted in adjusted polar coordinates (Burrell and Chisham, 2018).

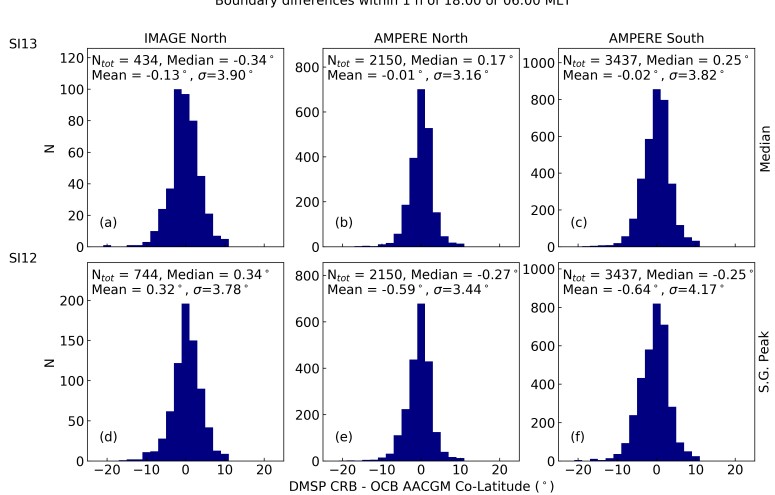

**Figure 5.** Histograms showing the differences between DMSP CRB and IMAGE or AMPERE OCB using paired boundaries that occur within 1 hr of 06:00 MLT or 18:00 MLT.

Figure 4 shows the distribution of CRB observations for the different DMSP satellites, OCB sources, and hemispheres. As was done with the DMSP SSJ observations, two years of CRBs and OCBs were paired in time after removing unreliable boundaries (as discussed in section 2). Note that both IMAGE and both AMPERE hemispheres show a similar spread of CRBs at different magnetic local times, with larger spreads near magnetic noon and midnight.

Figure 5 shows the histograms of the latitude differences between the DMSP CRBs and the IMAGE (panel a and d) or AMPERE (panels b, c, e, and f) OCBs. This figure shows the results for the median ellipse correction to obtain the AMPERE OCB in the top row and the S.G. peak ellipse correction in the bottom row. For the IMAGE histograms, panel (a) shows the results for the SI13 instrument and panel (d) shows the results for the SI12 instrument. In all cases the means and medians of the difference distributions behave similarly: most points lie within $1°$ of each other and the standard deviation of the distributions

is below $5°$ in all places. Additionally, the CRB is approximately collocated with both the AMPERE and IMAGE OCBs. This close agreement with the DMSP CRB and the similar behaviour of the IMAGE and AMPERE OCBs validates the AMPERE OCBs provided here.

## 5    Conclusions

This study modified traditional auroral boundary fitting methods to establish an MLT dependent relationship between the OCB

and the R1/R2 boundary. This was performed by determining the first moment of the distribution of differences between the R1/R2 boundary and the OCB (as measured by the DMSP SSJ instrument) for 1 hr MLT bins. These moments (which included





the median and the peak of a smoothed Gaussian) were then used to define the parameters of an elliptical function. This function specifies the distance between the OCB and R1/R2 boundary as a function of MLT.

The validity of this OCB, as well as previously determined IMAGE OCBs, were tested against the dawn and dusk measurements of the CRB (as measured by several DMSP IVM instruments). These boundaries were found to typically differ by less

than a degree.

As mentioned in the introduction, modeling and statistical studies in polar regions should avoid mixing measurements taken in the auroral oval and the polar cap. In combination, the AMPERE and IMAGE OCBs form the basis of a multi-solar cycle data set that could be used to improve high latitude statistical studies and climatological models. The data sets and software tools presented in this paper allow researchers to begin using adaptive, high latitude coordinates in their investigations.

*Code and data availability.* AMPERE data are available from the John Hopkins University Applied Physics Laboratory at http://ampere.jhuapl.edu/. We thank the AMPERE team and the AMPERE Science Center for providing the Iridium-derived data products. AMPERE boundaries can be requested from Steve Milan (steve.milan@leicester.ac.uk).

The IMAGE FUV data are provided courtesy of the instrument PI Stephen Mende (University of California, Berkeley). We thank the PI, the IMAGE mission, and the IMAGE FUV team for data usage and processing tools. The raw IMAGE data, and software, are

available from http://sprg.ssl.berkeley.edu/image/. The auroral boundary data set, and the methodology used to create it, can be found at https://www.bas.ac.uk/project/image-auroral-boundary-data/ or Chisham (2017a).

DMSP data are available at https://cedar.openmadrigal.org and https://cdaweb.gsfc.nasa.gov. DMSP SSJ boundaries may be obtained using the software available https://github.com/lkilcommons/ssj_auroral_boundary. DMSP CRBs can be requested from Yun-Ju Chen (yxc126130@utdallas.edu).

The software that was used to perform adaptive, high-latitude gridding can be found at https://github.com/aburrell/ocbpy or Burrell and

Chisham (2018).

*Author contributions.* AGB developed the concept, performed the data analysis, and wrote the manuscript. GC supported the conceptual development, provided feedback on the data analysis, and edited the manuscript. SEM provided the AMPERE R1/R2 boundaries and guidelines for their use, provided feedback on the conceptual development, and contributed to writing the manuscript. LK provided guidelines for the use of the DMSP SSJ boundaries, feedback on the validation, and edited the manuscript. Y-JC provided the DMSP CRBs, provided guide-

lines for their use in validation, and edited the manuscript. EGT provided feedback on the data validation efforts and edited the manuscript. BA is the PI of AMPERE.

*Competing interests.* No competing interests are present.

*Acknowledgements.* AGB is supported by the United States Chief of Naval Research (US CNR). GC is supported by United Kingdom Research and Innovation (UKRI) as part of the British Antarctic Survey Polar Science for Planet Earth Programme. LK is supported by



AFOSR Award No: FA9550-17-1-0258. YJC is supported by AFOSR MURI grant FA9559-16-1-0364 to University of Texas at Dallas. EGT is supported by NSF grant OPP-1836426.



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
