# Peer review of "AMPERE Polar Cap Boundaries"

_Annales Geophysicae, 2019_

## Short Comment (SC1) · 27 Aug 2019

Hi, The definitive reference for AMPERE science team data that includes a detailed description of the latest AMPERE data products and processing is as follows;

Waters, C.L., B.J. Anderson, D.L. Green, H. Korth, R.J. Barnes, and H. Vanhamaki, Science data products for AMPERE, in 'Ionospheric Multi-Spacecraft Analysis Tools', eds. M. Dunlop and H. Luhr, Springer, in press 2019.

[Figure]

---

## Author Comment (AC1) · 3 Sep 2019

Congratulations on your paper in press, I look forward to reading it. Of course, if and when this manuscript is published your paper will no longer be a *definitive* reference since the AMPERE polar cap boundaries will be missing!

Given that the AMPERE products used in this paper are all appropriately referenced and since I do not have access to your soon-to-be published paper, it is unlikely that it will be referenced in this manuscript. However, if you would like to send me a pre-print I would be happy to read it now and consider whether or not it would be appropriate to include.
* * *

---

## Referee Comment (RC1) · Anonymous Referee #1 · 16 Sep 2019

This study proposes to use data on field-aligned currents (FACs) inferred from magnetometer measurements on the IRIDIUM satellites to monitor the location of the open-closed boundary (OCB) of the Earth's magnetic field. Traditionally, the OCB has been determined from satellite measurements of precipitating electron fluxes. The authors relate the location of the area separating the so called Region 1 and Region 2 FAC currents (R1/R2) with the OCB location inferred from DMSP satellite measurements. The offset between these areas is assessed statistically for both hemispheres and in various MLT sectors, and the authors create a general equation to optimally describe it. To validate the technique, the authors compare OCBs inferred from R1/R2 data with the convection reversal boundary (CRB) inferred from DMSP ion drift measurements and with IMAGE optical data. All the data sets agree reasonably well at dawn and dusk. It is believed that the new boundaries based on R1/R2 data would allow researchers "to improve high latitude statistical studies and climatological models."

[Figure]

The paper is certainly a contribution to the field and I would recommend it for publication. However, the presentation requires significant improvement. It is important for the paper to be understandable to an un-prepared reader. The present version is not up to this expectation. My concerns are that the Introduction is too patchy to understand the justification for the study and there is no discussion of the significance of the work done in terms of future applications.

Generally, one can expect that the offset between the OCB and the R1/R2 separation line depends on the IMF intensity and orientation. In various parts of the manuscript, the explanations assume that there is the Dungey cycle going on, which is OK when the IMF Bz<0. However, some (unspecified) comparisons were done when the IMF Bz>0 (page 4, line 3). Thus, there is a sort of inconsistency in the explanations and conditions for which the comparison was actually done. Ideally, in my view, it would be important to establish that the offsets are IMF independent. Discussion of this aspect would be beneficial.

The paper needs further discussion the relationship between OCB and CRB. The latter was introduced without any explanations/comments at all. While OCBs and CRBs collocate well at dawn and dusk, they are quite off in closer to midnight sectors, see Fig. 4. What does this mean? Would this result "validate" the newly-introduced way of finding the OCB from R1/R2 data? This kind of little details urges the question on the value of the work done. One would expect these issues to be addressed in the Discussion section of the paper.

Below I list technical comments that I recommend the authors to think about.

1) p2: The first statement is supposed to be with respect to the ionosphere, not atmosphere. Then atmospheric motions and their effects can be introduced. In fact, abstract has more proper words than Introduction.

2) p.2: I am concerned about the usage of words: ... plasma drifts ...travel....

[Figure]

3) p.2: In lines 13-15, Joule heating is mentioned without connection to the previous narration on the plasma driven by the Dungey reconnection processes.

4) p.2: In lines 16-17 you state: "Due to the differences in ionospheric and thermospheric behavior in the auroral oval and the polar cap". It is not clear what it meant here, what kind of differences, they have not been introduced earlier.

5) p.2: In line 19 you mention "improved statistical and climatological results". Specifics need to be given here.

6) p.2: In line 20, the statement about OCB comes suddenly into the play, disconnected from the previous narration.

7) p.2, lines 26-27: Because the location of the Birkeland current system is tied to the OCB . . .. A reference or explanation is required here.

8) p.2, line 29: I would replace "measured by" by "inferred from"

9) p.2, line 34: Convection reversal boundary (CRB) is mentioned for the first time, but how it is related to the OCB and R1/R2 currents has never been mentioned.

10) p.3, line 21: "state of the magnetic field lines" – what is this?

11) p.3, lines 22-23: this statement should be in Introduction

12) p.9, line 3: Remove "R.H."

13) p.9, line 11: Remove dash

14) p.9, line 23: revise the statement

15) p.9: Fig 4 caption: . . .data show

16) p.15, line 35. . . ..Borovsky J.A. and Young T.D.

17) p.16, line 31: remove two "and"

18) p.16: Is reference on Jones correct?

19) p.16: Please, correct reference for Spiro

20) p.16: Reference on Zhu is incomplete

21) p.16: Many citations are not consistent with Ann. Geo. requirements in terms of style.
* * *

---

## Author Comment (AC2) · 9 Oct 2019

The authors would like to thank the reviewer for their thoughtful response. These changes will make the manuscript more accessible, improving its clarity and impact. We have addressed the reviewers comments as detailed below.

**1   General Comments**

With respect to the reviewer's concerns about the IMF, the confusion seems to arise from the discussion of the CRB detection algorithm. In an attempt to clear up this confusion, we have expanded the discussion in Section 2.2 to better discuss the algorithmic biases and how they affect the validation data set. In Section 4 we added: 'These local

times were chosen due to the MLT-dependent variations in the CRB-OCB relationship discussed in Section 2.2. Recall, as well, that no specific selection was made for IMF conditions. All IMF clock angles and magnitudes are considered together, as the AM-PERE OCBs will be valid at all IMF conditions when the OCB can be represented (to first order) by an ellipse.'

We decided not to break up the validation by clock angle, IMF steadiness, or IMF magnitude for several reasons. Firstly, at dawn and dusk the CRB-OCB relationship is not strongly dependent on the IMF (though it is at other MLTs). However, any dependence of the CRB-OCB relationship on IMF at this time will confuse the interpretation of the validation. Thus, it is most appropriate to consider all IMF conditions together and not attempt to infer if variations in the distribution are due to an IMF dependence on the part of the CRB or the AMPERE/IMAGE OCBs. Secondly, the number of points available as the data set is further broken down makes the results less statistically significant. However, the reviewer may be interested in seeing the figures that led the authors to come to this decision (Figures 1-5).

With respect to the reviewer's concerns about the relationship between the OCB and CRB, these concerns were addressed in Section 2.2. Specifically "Near magnetic noon and midnight, the flows tend to be mostly sunward or antisunward, meaning there is no clear reversal in the convection as a function of magnetic latitude". This, along with the other enumerated points in this section, make it clear that it is impossible for the CRB to be used in any sort of validation apart from the magnetic local times near dawn and dusk. The authors thought it was most appropriate to discuss this in the data selection portion of the paper, since these considerations were used to select an appropriate validation data set. However, to ensure that reader recalls the details of this discussion when the validation is brought up, we have added this sentence to the validation Section: 'These local times were chosen due to the MLT-dependent variations in the CRB-OCB relationship discussed in Section 2.2.'. In addition, we have expanded the discussion of the CRB in the Introduction.

**2 Technical Comments**

These points refer to the numbers of the technical comments made in RC1.

1. We changed the wording in the introduction to be more similar to that used in the abstract.

2. Revised wording in the introduction.

3. Removed the Joule Heating example.

4. Clarified this statement to read: 'Due to these and other differences in MIT coupling processes in the auroral oval and the polar cap, it is desirable to have a coordinate system that indicates in which region measurements were taken.'

5. We disagree with the reviewer that specifics were not provided in this sentence, as this phrase immediately follows and refers to three peer-reviewed journal articles that demonstrate the improvements that can be made in statistical and climatological studies by using OCB oriented coordinates. However, to avoid confusion we have added a specific example from one of these articles: '(for example, Chisham (2017) demonstrated the difference between using magnetic and OCB oriented coordinates when studying the climatological behaviour of the plasma drift vorticity)'

6. Changed the introduction to introduce the OCB by name in the second paragraph.

7. Changed the wording to be more specific and added a reference to the review paper by Coxon et al. (2018). 'Because the location of the Birkeland current system is tied to the expansion and contraction of the polar cap under quiescent and disturbed conditions (Coxon, et al., 2018).'

8. Replaced 'measured by' with 'inferred from particle precipitation measurements made by'

9. The CRB is now introduced in the third paragraph in the introduction, and related to the Dungey cycle (which is used as a reference point for all of the other examples).

10. Clarified text to say: 'Because the direction of convective plasma drifts are strongly tied to the motion and state (i.e., open or closed) of the magnetic field lines'

11. Moved to the introduction.

12. Fixed author name order in bibTeX (here and elsewhere)

13. Removed dash in reference year

14. The statement was revised to be: 'The similarity between the two fits can be quantified by comparing the differences between $a_{Median}$ and $a_{S.G.\ Peak}$ (0.40°) and the typical difference between the hourly median and S.G. peak values (0.49°); the differences between the eccentricity and angular offset are even less significant.'

15. Fixed as suggested

16. Fixed editor names.

17. Removed the two extra 'and's in the article title.

18. The Jones citation is correct (more correct with the dashed year), as it is obtained from the SciPy.org citation guide available at: https://www.scipy.org/citing.html

19. Fixed title in Spiro reference.

20. Updated the Zhu reference.

21. Reviewed all bibTeX entries, removing unneeded fields that may have caused the Copernicus template to create non-standard looking references.
* * *
[Figure]

IMF coverage of CRBs within 1 h of 18:00 or 06:00 MLT

IMAGE

North, SI13
$N_{tot}$=434
$N_{max}$=52

AMPERE

North
$N_{tot}$=2150
$N_{max}$=285

North, SI12
$N_{tot}$=744
$N_{max}$=102

South
$N_{tot}$=3437
$N_{max}$=438

Clock Angle (∘)

Clock Angle (∘)

$B_{YZ}$ Mag (nT)

$B_{YZ}$ Mag (nT)

$N_{max}$

N

**Fig. 1.** IMF coverage by clock angle and magnitude

Boundary differences within 1 h of 18:00 or 06:00 MLT and -45$^\circ$ < $\theta$ < 45$^\circ$

SI13

IMAGE North — $N_{tot}$ = 33, Median = -0.42$^\circ$, Mean = 0.18$^\circ$, $\sigma$=3.44$^\circ$ (a)

AMPERE North — $N_{tot}$ = 302, Median = -0.38$^\circ$, Mean = -0.71$^\circ$, $\sigma$=3.68$^\circ$ (b)

AMPERE South — $N_{tot}$ = 424, Median = -0.12$^\circ$, Mean = -0.38$^\circ$, $\sigma$=4.11$^\circ$ (c)

Median

SI12

$N_{tot}$ = 79, Median = -0.26$^\circ$, Mean = 0.21$^\circ$, $\sigma$=3.61$^\circ$ (d)

$N_{tot}$ = 302, Median = -1.04$^\circ$, Mean = -1.51$^\circ$, $\sigma$=4.07$^\circ$ (e)

$N_{tot}$ = 424, Median = -0.68$^\circ$, Mean = -1.05$^\circ$, $\sigma$=4.46$^\circ$ (f)

S.G. Peak

DMSP CRB - OCB AACGM Co-Latitude ($^\circ$)

**Fig. 2.** +Bz Validation

Boundary differences within 1 h of 18:00 or 06:00 MLT and $135° < θ < 225°$

SI13

IMAGE North

$N_{tot} = 149$, Median = $0.04°$
Mean = $-0.11°$, $σ=3.40°$

(a)

AMPERE North

$N_{tot} = 565$, Median = $0.89°$
Mean = $0.86°$, $σ=2.23°$

(b)

AMPERE South

$N_{tot} = 781$, Median = $0.85°$
Mean = $0.58°$, $σ=2.85°$

(c)

Median

SI12

$N_{tot} = 223$, Median = $0.67°$
Mean = $0.58°$, $σ=3.41°$

(d)

$N_{tot} = 565$, Median = $0.50°$
Mean = $0.41°$, $σ=2.36°$

(e)

$N_{tot} = 781$, Median = $0.42°$
Mean = $0.07°$, $σ=3.04°$

(f)

S.G. Peak

DMSP CRB - OCB AACGM Co-Latitude ($°$)

**Fig. 3.** -Bz Validation

Boundary differences within 1 h of 18:00 or 06:00 MLT and $225° < \theta < 315°$

SI13

**IMAGE North**

(a) $N_{tot} = 142$, Median $= -0.64°$
Mean $= -0.29°$, $\sigma = 3.54°$

**AMPERE North**

(b) $N_{tot} = 736$, Median $= -0.25°$
Mean $= -0.34°$, $\sigma = 3.42°$

**AMPERE South**

(c) $N_{tot} = 1051$, Median $= 0.76°$
Mean $= 0.43°$, $\sigma = 3.84°$

Median

SI12

(d) $N_{tot} = 262$, Median $= 0.37°$
Mean $= 0.24°$, $\sigma = 3.23°$

(e) $N_{tot} = 736$, Median $= -0.81°$
Mean $= -1.00°$, $\sigma = 3.76°$

(f) $N_{tot} = 1051$, Median $= 0.32°$
Mean $= -0.12°$, $\sigma = 4.21°$

S.G. Peak

DMSP CRB - OCB AACGM Co-Latitude (°)

**Fig. 4.** -By Validation

Boundary differences within 1 h of 18:00 or 06:00 MLT and $45° < \theta < 135°$

SI13

IMAGE North

$N_{tot} = 87$, Median $= -0.31°$
Mean $= 0.13°$, $\sigma=5.36°$

(a)

AMPERE North

$N_{tot} = 399$, Median $= -0.11°$
Mean $= -0.18°$, $\sigma=3.20°$

(b)

AMPERE South

$N_{tot} = 963$, Median $= -1.23°$
Mean $= -0.91°$, $\sigma=4.23°$

(c)

Median

SI12

$N_{tot} = 147$, Median $= 0.14°$
Mean $= 0.13°$, $\sigma=5.03°$

(d)

$N_{tot} = 399$, Median $= -0.51°$
Mean $= -0.65°$, $\sigma=3.41°$

(e)

$N_{tot} = 963$, Median $= -1.96°$
Mean $= -1.68°$, $\sigma=4.62°$

(f)

S.G. Peak

DMSP CRB - OCB AACGM Co-Latitude ($°$)

**Fig. 5.** +By Validation

---

## Referee Comment (RC2) · Anonymous Referee #2 · 24 Oct 2019

This paper looks at the open closed magnetic field line boundaries (OCB) of Earth's magnetosphere. It provides a new set of OCBs, derived from Active Magnetosphere and Planetary Electrodynamics Response Experiment (AMPERE) Region 1 and Region 2 (R1/R2) field aligned current boundaries, which are then compared to older measurements of the OCB from the Defense Meteorological Satellite Programme's (DMSP) electron flux measurements obtained with the Special Sensor J (SSJ), as well as measurements obtained by the Imager for Magnetopause-to-Aurora Global Exploration (IMAGE) satellite. A parameterisation is then performed on the AMPERE R1/R2 boundary measurements, which are initially only taken at dusk & dawn, to obtain an oval fit for the OCB offset as a function of magnetic local time dependence. The measurements and comparison of different datasets show that there is little offset (generally within $\sim 1$ degree) between the R1/R2 boundary measurements and the DMSP OCB measurements.

[Figure]

General comments: The comparisons in this paper provide a basis for a new adaptive framework for analysing data in the future and the large-scale analysis provides confidence and validation for this method. Overall, this adds to a large body of existing science and thus will be relevant to the research community. I believe that this paper is worthy of publication, but some clarification is necessary first (with some additional analysis being potentially necessary) and some further discussion of the results would be highly desirable (listed below).

Specific comments (Major):

- Section 2.1: How do you identify boundaries using AMPERE? You talk about AMPERE but not how the boundaries are specifically identified, but you have already mentioned this in passing. In the next section, you go straight into the OCB determination and it would be good to have the same here for consistency.

- Section 2.2, approx. L 25+: What about the fact that the polar cap may move towards nightside? I.e. dusk-dawn measurements may not always be comparable with each other and this may create some inconsistencies. Is this going to add an error to your estimations? This is something which is mentioned later, but I think this needs to be addressed straight away.

- Section 3, L 11: Your peak-to-peak magnitude will somewhat depend on the latitude and the offset between the two peaks also. Is this considered? If not, you may have to normalise the peak-to-peak ratio. E.g. for a circle closer to the pole, the points will be closer spaced than further towards the equator, so more points will lie in each AMPERE MLT than closer to the equator, which could skew the results if not taken into account.

- Section 3, L 16: What is the justification for the 10 min timescale? Is this not too long? For example, Substorm contractions of the OCB for example, can occur on timescales shorter than this.

- Figure 1: Can you plot om the median/quartiles of the DMSP boundaries also? There are a lot of overlapping points, so it's hard to see.

- Page 8, L 2: Why is one number negative and one positive? Is this due to inter-hemispheric differences? In general, it would add a lot if you could add a discussion of interhemispheric differences and asymmetries in the context of existing research.

- Page 9, L 6: Why fit an ellipse now? Why not fit an ellipse straight away? I.e why aren't you fitting one to the AMPERE data?

- Page 9, L 11: Is there a version number that goes with this software?

- Page 10, L 11: Again, is the 10 min timescale not too long?

Technical corrections (Minor):

- Abstract L 15: insert "the Imager for " before "Magnetopause-to-Aurora Global Exploration".

- Section 1, L 2: add "and the solar wind" after "between the ionosphere and magnetosphere".

- Page 9, L 11: Why is there a dash after the Jones reference?

- Page 10, L 11: Which OCBs? There is a lot mentioning of different OCBs in the previous sentences, so it's unclear which one you mean in this sentence.

---

## Author Comment (AC3) · 13 Nov 2019

The authors would like to thank the reviewer for their response. We have addressed the reviewers comments as detailed below.

**1   Major Comments**

1. (Section 2.1) We moved the discussion of the R1/R2 FAC current boundaries from Section 3 to Section 2.1.

2. (Section 2.2) As the reviewer notes, this is discussed later in the paper when the data is used. We do not believe it makes sense to include it here, since we are discussing only the CRBs in this section and not the pairing and comparison.

[Figure]

3. (Section 3, L11) This portion of the paper is presenting a well established data set, as noted on p4, Line 26. We refer the reviewer to Milan et al. (2015) for a detailed answer to this question, as all of these concerns were considered when this method was developed.

4. (Section 3, L16) The justification for the 10 minute timescale has to do with the AMPERE processing. We refer the reviewer back to Section 2.1, which states that the AMPERE FAC patters are calculated from 10 minute averages. However, not much of the data has time differences of 10 minutes. Figure 1 in this response shows the histograms of the time differences for DMSP and AMPERE pairs in each hemisphere. To allay the concerns of any readers, we have added the following statements to the text: *The 10 min window for pairing boundaries was chosen because of the 10 min averaging performed on the AMPERE FAC maps (see Section 2.1). However, over 90% of northern hemisphere pairs and over 80% of southern hemisphere pairs have a temporal difference of 1 min or less.*

5. (Figure 1). We experimented with several visualisions for this figure. Adding the medians/quartiles of the DMSP boundaries made the figure too busy unless the scatter points were removed. However, removing the scatter points also removed the information about the limits of the satellite boundaries. In the interest of providing a clear visual representation, we prefer to leave the figure as is. Especially since Figure 2 and Table 1 provide detailed hourly data about the median paired differences.

6. (P8, L2) As stated in lines 1 and 2 on page 8, the mean difference between the northern and southern MLT medians (when both hemispheres have data) is $-0.3°$ and the mean difference between the northern and southern MLT smoothed Gaussian peaks (when both hemispheres have data) is $0.23°$. This is related to the differences in the statistics rather than a hemispheric asymmetry. In fact, it shows that there is no significant interhemispheric asymmetry between the

DMSP SSJ and AMPERE R1/R2 FAC boundary differences. This is stated on the next line: *This difference is small enough to justify combining the northern and southern hemispheric $\Delta\phi$, since it is much smaller than the mean standard deviation of the MLT distributions ($\bar{\sigma} = 2.66°$ for the overlapping MLT bins).*

7. (P9, L6) We refer the reviewer to Milan et al. (2015) for the reasons behind fitting a circle to the AMPERE data.

8. (P9, L11) Added the SciPy version number to the reference.

9. (P10, L11) We refer the reviewer back to Figure 1 in this response.

**2   Minor Comments**

1. (Abstract L15) Added.

2. (Section 1, L2) This paragraph was changed at the request of Reviewer 1, and this sentence was removed.

3. (P9, L11) Because this is the standard reference provided by SciPy. However, we have removed this dash as requested.

4. (P10, L11) When "OCBs" is used with no qualifier, it applies to all OCBs. Every instance that refers to a specific OCB is prefaced by either 'AMPERE' or 'IMAGE'.
* * *
[Figure]

[Figure]

**Fig. 1.** Time difference in seconds between paired DMSP SSJ and AMPERE R1/R2 FAC boundaries.

---

## Author Response (AR1)

The authors would like to thank both reviewers for their responses, and for agreeing to evaluate the revised manuscript. Point-by-point responses to each reviewer are included below (Section 1), followed by a list of all relevant changes (Section 2), and a marked-up version of the manuscript. We hope that this will aid the reviewers in their evaluation of the revised manuscript.

**1 Reveiwer responses**

**1.1 Reviewer 1 General Comments**

With respect to the reviewer's concerns about the IMF, the confusion seems to arise from the discussion of the CRB detection algorithm. In an attempt to clear up this confusion, we have expanded the discussion in Section 2.2 to better discuss the algorithmic biases and how they affect the validation data set (page 5, lines 11-14 in the revised manuscript). In Section 4 (p 11, lines 3-6 in the revised manuscript) we added: 'These local times were chosen due to the MLT-dependent variations in the CRB-OCB relationship discussed in Section 2.2. Recall, as well, that no specific selection was made for IMF conditions. All IMF clock angles and magnitudes are considered together, as the AMPERE OCBs will be valid at all IMF conditions when the OCB can be represented (to first order) by an ellipse.'

We decided not to break up the validation by clock angle, IMF steadiness, or IMF magnitude for several reasons. Firstly, at dawn and dusk the CRB-OCB relationship is not strongly dependent on the IMF (though it is at other MLTs). However, any dependence of the CRB-OCB relationship on IMF at this time will confuse the interpretation of the validation. Thus, it is most appropriate to consider all IMF conditions together and not attempt to infer if variations in the distribution are due to an IMF dependence on the part of the CRB or the AMPERE/IMAGE OCBs. Secondly, the number of points available as the data set is further broken down makes the results less statistically significant. The figures that led to these conclusions are available in the Author's public response to the reviewers.

With respect to the reviewer's concerns about the relationship between the OCB and CRB, these concerns were addressed in Section 2.2. Specifically (p 5, lines1-2 in the revised manuscript) "Near magnetic noon and midnight, the flows tend to be mostly sunward or antisunward, meaning there is no clear reversal in the convection as a function of magnetic latitude". This, along with the other enumerated points in this section, make it clear that it is impossible for the CRB to be used in any sort of validation apart from the magnetic local times near dawn and dusk. The authors thought it was most appropriate to discuss this in the data selection portion of the paper, since these considerations were used to select an appropriate validation data set. However, to ensure that reader recalls the details of this discussion when the validation is brought up, we have added this sentence to the validation Section (p 11, lines 3-4 in the revised manuscript): "These local times were chosen due to the MLT-dependent

variations in the CRB-OCB relationship discussed in Section 2.2.'. In addition, we have expanded the discussion of the CRB in the Introduction.

**1.2** Reviewer 1 Technical Comments**

These points refer to the numbers of the technical comments made in RC1.

- 1. We changed the wording in the introduction to be more similar to that used in the abstract (p 2, lines 1-2 in the revised manuscript).
- 2. Revised wording in the introduction (pages 2-3 in the revised manuscript).
- 3. Removed the Joule Heating example (page 3, lines 2-3 in the marked-up manuscript).
- 4. Clarified this statement (page 2, lines 29-30 in the revised manuscript) to read: 'Due to these and other differences in MIT coupling processes in the auroral oval and the polar cap, it is desirable to have a coordinate system that indicates in which region measurements were taken.'
- 5. We disagree with the reviewer that specifics were not provided in this sentence, as this phrase immediately follows and refers to three peer-reviewed journal articles that demonstrate the improvements that can be made in statistical and climatological studies by using OCB oriented coordinates. However, to avoid confusion we have added a specific example from one of these articles (p 2 lines 32-33 in the revised manuscript): '(for example, Chisham (2017) demonstrated the difference between using magnetic and OCB oriented coordinates when studying the climatological behaviour of the plasma drift vorticity)'
- 6. Changed the introduction to introduce the OCB by name in the second paragraph (p 2, line 13 in the revised manuscript).
- 7. Changed the wording to be more specific and added a reference to the review paper by Coxon et al. (2018). 'Because the location of the Birkeland current system is tied to the expansion and contraction of the polar cap under quiescent and disturbed conditions (Coxon, et al., 2018, and references therein),...' (p 3, lines 6-8 in the revised manuscript)
- 8. Replaced 'measured by' with 'inferred from particle precipitation measurements made by' on p 3 line 10 in the revised manuscript.
- 9. The CRB is now introduced in the third paragraph in the introduction (p 2, line 25 in the revised manuscript), and related to the Dungey cycle (which is used as a reference point for all of the other examples).
- 10. Clarified text to say: 'Because the direction of convective plasma drifts are strongly tied to the motion and state (i.e., open or closed) of the magnetic field lines' on p 3 lines 18-19 in the revised manuscript.

- 11. Moved to the introduction (p 3 lines 18-20 in the revised manuscript).
- 12. Fixed author name order in bibTeX (here and elsewhere).
- 13. Removed dash in reference year (p 10, line 11 in the revised manuscript).
- 14. The statement was revised to be: 'The similarity between the two fits can be quantified by comparing the differences between  $a_{Median}$  and  $a_{S.G. Peak}$  $(0.40^{\circ})$  and the typical difference between the hourly median and S.G. peak values  $(0.49^{\circ})$ ; the differences between the eccentricity and angular offset are even less significant.' (p 10, lines 22-24 in the revised manuscript).
- 15. Fixed as suggested (p 12 line 1 in the revised manuscript).
- 16. Fixed editor names (p 16, line 2 in the revised manuscript).
- 17. Removed the two extra 'and's in the article title (p 16, line 22 in the revised manuscript).
- 18. The Jones citation is correct (more correct with the dashed year), as it is obtained from the SciPy.org citation guide available at: https://www.scipy.org/citing.html
- 19. Fixed title in Spiro reference (p 16, line 31 in the revised manuscript).
- 20. Updated the Zhu reference (p 16, line 35 in the revised manuscript).
- 21. Reviewed all bibTeX entries, removing unneeded fields that may have caused the Copernicus template to create non-standard looking references (p 15 and 16 in the revised manuscript).

**1.3** Reviewer 2 Major Comments**

- 1. (Section 2.1) We moved the discussion of the R1/R2 FAC current boundaries from Section 3 to Section 2.1 (p 4 lines 1-18 in the revised manuscript).
- 2. (Section 2.2) As the reviewer notes, this is discussed later in the paper when the data is used. We do not believe it makes sense to include it here, since we are discussing only the CRBs in this section and not the pairing and comparison.
- 3. (Section 3, L11) This portion of the paper is presenting a well established data set, as noted on p4, Line 26 (p4, line 1 in the revised manuscript). We refer the reviewer to Milan et al. (2015) for a detailed answer to this question, as all of these concerns were considered when this method was developed.
- 4. (Section 3, L16) The justification for the 10 minute timescale has to do with the AMPERE processing. We refer the reviewer back to Section 2.1, which states that the AMPERE FAC patters are calculated from 10

minute averages. However, not much of the data has time differences of 10 minutes (p 3, line 31 in the revised manuscript). The figure included in the public response shows the histograms of the time differences for DMSP and AMPERE pairs in each hemisphere. To allay the concerns of any readers, we have added the following statements to the text: The 10 min window for pairing boundaries was chosen because of the 10 min averaging performed on the AMPERE FAC maps (see Section 2.1). However, over 90% of northern hemisphere pairs and over 80% of southern hemisphere pairs have a temporal difference of 1 min or less. (p 6, line 9 of the revised manuscript).

- 5. (Figure 1). We experimented with several visualisions for this figure. Adding the medians/quartiles of the DMSP boundaries made the figure too busy unless the scatter points were removed. However, removing the scatter points also removed the information about the limits of the satellite boundaries. In the interest of providing a clear visual representation, we prefer to leave the figure as is. Especially since Figure 2 and Table 1 provide detailed hourly data about the median paired differences.
- 6. (P8, L2) As stated in lines 1 and 2 on page 8 (p 6, lines 28-29 in the revised manuscript), the mean difference between the northern and southern MLT medians (when both hemispheres have data) is -0.3° and the mean difference between the northern and southern MLT smoothed Gaussian peaks (when both hemispheres have data) is 0.23°. This is related to the differences in the statistics rather than a hemispheric asymmetry. In fact, it shows that there is no significant interhemispheric asymmetry between the DMSP SSJ and AMPERE R1/R2 FAC boundary differences. This is stated in the following sentence (p 6 lines 29-31 in the revised manuscript): This difference is small enough to justify combining the northern and southern hemispheric  $\Delta \phi$ , since it is much smaller than the mean standard deviation of the MLT distributions ( $\bar{\sigma} = 2.66^{\circ}$  for the overlapping MLT bins).
- 7. (P9, L6) We refer the reviewer to Milan et al. (2015) for the reasons behind fitting a circle to the AMPERE data.
- 8. (P9, L11) Added the SciPy version number to the reference (p 16, line 7 in the revised manuscript).
- 9. (P10, L11) We refer the reviewer back to point 4 in this list.

**1.4 Reviewer 2 Minor Comments**

- 1. (Abstract L15) Added (p 1, line 15 in the revised manuscript).
- 2. (Section 1, L2) This paragraph was changed at the request of Reviewer 1, and this sentence was removed.

- 3. (P9, L11) Because this is the standard reference provided by SciPy. However, we have removed this dash as requested (p 10, line 11 in the revised manuscript).
- 4. (P10, L11) When "OCBs" is used with no qualifier, it applies to all OCBs. Every instance that refers to a specific OCB is prefaced by either 'AM-PERE' or 'IMAGE'.

**2 List of all relevant changes**

All page (P) and line (L) numbers in the following list refer to the marked-up manuscript.

- 1. (P1, L15): Fixed the name of the IMAGE mission.
- 2. (P2-3): Rewrote introduction, reorganising paragraphs to introduce the most important elements earlier, reducing the number of examples, and spending more time on the examples that are included.
- 3. (P4, L 11) used more explicit wording to make the sentence easier to understand.
- 4. (P4, L 12-29 and P6, L15 P7, L2) Reorganized data in Sections 3 and 2.1, to introduce the R1/R2 FAC boundary fitting method earlier.
- 5. (P5, L7-11) Rewrote paragraph to be consistent with the newly rewritten introduction.
- 6. (P5, L24-27) Added sentences to clarify the limitations and resulting coverage gaps in the validation data set.
- 7. (P7, L3) Rewrote sentence to reflect new section organisation.
- 8. (P7, L4-7) Added sentences explaining reasoning behind the temporal pairing window, and providing details about the typical time difference of these pairs.
- 9. (P11, L7-10) rewrote sentence to be more clear.
- 10. (P12, L18-21) Added sentences to remind the reader what was previously discussed in Section 2.
- 11. (P13, Figure 4 caption) Fixed a grammatical mistake pointed out by Reviewer 1.
- 12. (P14, L1-2) Clarified text in sentence discussing Figure 5.
- 13. (P14, L14) Fixed grammar in this sentence.
- 14. (P15, L9) Updated data availability of the AMPERE R1/R2 FAC boundaries.

- 15. (P17, L12 (numbered)) Fixed editors.
- 16. (P17, L14 (numbered)) Fixed bibtex format for author names.
- 17. (P17, L17 (numbered)) Fixed SciPy reference (year and version number).
- 18. (P17, L376 (numbered)) Fixed title.
- 19. (P18) Updated reference.

**AMPERE Polar Cap Boundaries**

Angeline G. Burrell1, Gareth Chisham2, Stephen E. Milan3, Liam Kilcommons4, Yun-Ju Chen5, Evan G. Thomas6, and Brian Anderson7 1U.
[revised manuscript text omitted]

---

## Author Response (AR2)

The authors would like to thank both reviewers for their responses.

**1   Reveiwer responses**

**1.1   Reviewer 1**

We would like to thank this reviewer for taking the time to carefully read the manuscript and consider our responses. Their clear feedback greatly improved the manuscript.

**1.2   Reviewer 2**

We are happy to further clarify our responses, as noted below. In these responses, we have referred to both reviews provided by Reviewer 2, as the second review does not stand on its own. Due to the incompleteness of the concerns raised in the second review, we apologise in advance for any misunderstandings present in this response, as well as any repetition in our responses.

2. *CRB Selection:* The reviewer appears to have misunderstood much of Section 4 and Figure 4. Reading both their original concern and this secondary concern, we attempt to answer all points here:

   - *What about the fact that the polar cap may move towards nightside? I.e. dusk-dawn measurements may not always be comparable with each other and this may create some inconsistencies.* The CRB locations are converted into OCB coordinates and the MLT selection occurs within this coordinate system. Thus, the motion of the polar cap has been treated consistently, and will not add error to the validation.

   - *The choice of using only values along the dusk-dawn meridian for the CRB has implications for the rest of the analysis, which is why it is crucial to discuss it sooner in the paper.* The CRBs are only used as a validation data set, and have no implications in the determination of the AMPERE OCB proxy.
     The MLT selection of the CRBs does have implications for the validation: we have only validated the AMPERE OCB proxies near dawn and dusk. In the future, if another validation data set becomes available, we plan on validating both the IMAGE and AMPERE OCB proxies at other magnetic local times. However, at this time such a validation is not possible. This incomplete validation is still an improvement on past work, such as Chisham (2017), which provided no statistical validation of their OCB proxy.

   - *It could explain the large errors seen in Figure 4, which are not even discussed in the paper.* The large distribution of points in Figure 4 shows the CRBs from many IMF conditions and all MLT, not just

those deemed appropriate for validation. These were provided to back up our reasons for to implementing a limited MLT validation period. These reasons are listed in Section 2.2, and referenced again in Section 4. Thus, the reviewers point does not make sense. Figure 4 shows all coincident CRBs, not just those near dawn and dusk. If it only showed CRBs near dawn and dusk, all other MLTs would not be present in Figure 4. To ensure no other readers make this mistake, we have added "Although all CRBs paired with IMAGE or AMPERE OCBs are shown here, only CRBs within 1 h of 06:00 or 18:00 MLT were used in this validation."

With respect to *large errors . . . not even discussed in the paper*, the so-called 'errors' are presented in Figure 5 and discussed extensively in Section 4.

3. (Section 3, L11) As the reviewer has noted, this manuscript describes some improvements that have been made since the original Milan et al. (2015) publication. We respectfully disagree that *Whilst the AMPERE data may be well established, the peak-to-peak ratio derived from it is not*, as this data set currently forms the basis for several papers and thus has been reviewed many times (e.g., Milan et al. (2017) doi:10.1007/s11214-017-0333-0, Milan et al. (2018) doi:10.1029/2018JA025645, Milan et al. (2019) doi:10.1029/2018JA025969). Moreover, since the data set was originally produced 5 years ago, it has been studied extensively by the authors and found to provide a good fit. One test that has been applied, but is yet unpublished (paper in preparation), is a comparison between a different FAC boundary identification method. The results of the two methods are both robust and consistent with each other.

However, we are happy to provide further clarification of the method. We note that this entire explanation is contained within the combination of Milan et al. (2015) reference and Section 2.1 of the revised manuscript.

The R1/R2 FAC boundary detection method seeks to find both the centre of the circle that best denotes this boundary and its radius, as these are both expected to vary as the polar cap expands, contracts, and shifts. This method begins by choosing an arbitrary centre location, and summing the currents. A peak-to-peak magnitude is then obtained. This is performed multiple times, changing the centre of the circle, until a maximum in the peak-to-peak magnitude is obtained. A maximum in the peak-to-peak magnitude indicates that one has found the ideal centre for a circular R1/R2 FAC boundary. The distance from the center to the inflection point gives the radius of that circle. The minimum possible peak-to-peak magnitude was chosen to eliminate times when the R1/R2 FAC systems could not be well described by a circular boundary, but also eliminates times when the FACs are weak.

We further address many of the individual 'examples' the reviewer provided to ensure that we have fully addressed this question:

– *for a circle closer to the pole, the points will be closer spaced than further towards the equator, so more points will lie in each AMPERE MLT than closer to the equator, which could skew the results if not taken into account.* If the circle is centred exactly at the geomagnetic pole, then because the points are distributed equally in azimuthal angle, exactly the same number of points will fall in each MLT sector, irrespective of the radius of the circle. As the centre of the circle moves away from the pole (because the auroral and FAC ovals are generally displaced a few degrees towards the nightside), then a small bias is introduced into the number of points that fall within each sector. However, this effect is negligible for three reasons:

1. the displacement of the centre of the circle from the pole is small with respect to the typical radius of the FAC ovals, so the bias is minimal;

2. the points sample evenly in circumference around the FAC ovals, and because AMPERE measures current density, this is exactly the behaviour the algorithm should have;

3. the FAC magnitudes are usually largest at dawn and dusk, and here the bias in sampling is at its least, as the circle is usually displaced along the noon-midnight meridian, not the dawn-dusk meridian.

– *Furthermore, if it was as well established a method as the authors claim, then why does the Milan et al. 2015 paper use 48 equally spaced points as opposed to the 200 points presented here?* When the algorithm was first implemented, the centre of the circle was assumed fixed, such that the same number of points always fell in the same MLT sectors. 48 points was chosen as it was shown that the MLT sectors were consistently sampled. When the algorithm was modified to relocate the centre to achieve the best fit, it was decided to increase the number of points to ensure that the circumference of the circle was as evenly sampled as possible with respect to the co-latitutde/MLT grid of the AMPERE data set. The number 200 was chosen empirically: different numbers were used during testing, with little change to the results of the fitting.

– *Page 4, line 26 reads "[something not relevant]"* This page and line reference refer to the original manuscript. In the response, all line and page references that were not followed by "in the revised manuscript" refer to the line and page numbers in the discussion manuscript. This was done to facilitate an open discussion, since those following the online discussion do not have access to the revised manuscript. The lines in the revised manuscript are P4, L1-5.

5. (Figure 1) *Could you make the lines indicating the MLT sectors longer/shorter depending to reflect the quartiles?* The length of the inner and outer lines

already mark the first and third quartiles (as stated in the Figure 1 caption), and are not symmetric.

7. *This does not adequately answer my questions. The Milan et al. 2015 paper simply states "we assume that the R1/R2 current regions are approximately circles centered on a point displaced a few degrees antisunward from the geomagnetic pole", which does not explain this further.* We are not fitting an ellipse to the AMPERE data because the Milan et al. (2015) method only works with a circle and because this boundary is well represented by a circle. This method is preferable to other methods (for example selecting MLT bins and fitting a circle or ellipse to these points) because it appropriately weights the regions with stronger FACs when performing the fit.

   Our goal in this project is to establish and include the MLT dependent relationship between the OCB and R1/R2 boundary based on the previous boundary fitting procedures using the simplest reasonable fit. We want to use the simplest reasonable fit because a proxy is never going to be as accurate as a measurement and should not present itself as such. We found that the best way to achieve this aim was to take the R1/R2 FAC boundary (a circle) and add the DMSP SSJ correction (an ellipse). This will take care of the deviation from the circle shape (if any) and also take care of the relationship between the OCB and R1/R2 boundaries as a function of MLT. Even with this simplicity, the resulting AMPERE OCB proxies are more complicated than previous OCB proxies (e.g., the IMAGE FUV proxies, which are circle fits). This is discussed in the revised manuscript on P 10 L 1-7.

**2   List of all relevant changes**

[revised manuscript text omitted]